# ADVERSARIAL TWIN NEURAL NETWORKS: MAXIMIZING PHYSICS RECOVERY FOR PHYSICAL SYSTEM

## ABSTRACT

The exact modeling of modern physical systems is challenging due to the expanding system territory and insufficient sensors. To tackle this problem, existing methods utilize sparse regression to find physical parameters or add another virtual learning model like a Neural Network (NN) to universally approximate the unobserved physical quantities. However, the two models can't perfectly play their own roles in joint learning without proper restrictions. Thus, we propose (1) sparsity regularization for the physical model and (2) physical superiority over the virtual model. They together define output boundaries for the physical and virtual models. Further, even the two models output properly, the joint model still can't guarantee learning maximal physical knowledge. For example, if the data of an observed node can linearly represent those of an unobserved node, these two nodes can be aggregated. Therefore, we propose (3) to seek the dissimilarity of physical and virtual outputs to obtain maximal physics. To achieve goals (1) − (3), we design a twin structure of the Physical Neural Network (PNN) and Virtual Neural Network (VNN), where sparse regularization and skip-connections are utilized to guarantee (1) and (2). Then, we propose an adversarial learning scheme to maximize output dissimilarity, achieving (3). We denote the model as the Adversarial Twin Neural Network (ATN). Finally, we conduct extensive experiments over various systems to demonstrate the best performance of ATN over other state-of-the-art methods.

## 1 INTRODUCTION

Internet of Everything (IoE) expands quickly to interconnect various devices. The systematic planning, modeling, and control of IoE can bring many benefits to society (Li et al., 2020). However, it remains an open question on how to efficiently model various grids with different levels of system information. For example, cyber-physical systems may be partially traceable by physical laws with limited sensing (Mulani et al., 2020). Such scenarios have a significant appearance nowadays on the grid edges of various physical systems due to the availability of low-cost, low-power sensor technology. These edge areas have unequal sensors but can be used to recover some physical laws completely or partially (Divan et al., 2014; Hu & Tang, 2020).

Learning physical equations from data is a central topic of Artificial Intelligence (Sahoo et al., 2018; Udrescu & Tegmark, 2020). While there are many related studies, e.g., symbolic regression (Petersen, 2019) and its variation with sparsity regularization (Brunton et al., 2016b), these methods are inapplicable to IoE or other complex physical systems due to the incomplete system observability. To tackle the issue, (Li & Weng, 2021) builds a shallow-deep structure to learn physics in a shallow neural network and simultaneously approximate the hidden components in a deep NN. Such a method, however, doesn't give a clear boundary between two NNs and suffers risks of imbalanced representation power between the physical and the virtual. Thus, there is a need to enforce restrictions for the model output within the boundary and keep the physical consistency.

Further, existing methods can hardly guarantee that the extracted physical model is operationally optimal. For example, in the realistic operation, if the data of an observed node can linearly represent those of an unobserved node, these two nodes can be aggregated, leading to the so-called Network Reduction (NR) (Oh, 2012; Zheng et al., 2021). On the other hand, some components of hidden quantities that can't be represented via the observed data can be treated as noise. Thus, our target is to identify an optimal reduced grid with operationally maximal physics. Namely, the reduced grid

should fully represent the input-output relationship of the observed measurements under a certain noise level.

Based on the above observations, we propose Adversarial Twin NNs (ATN) for optimal system modeling, where a Physical Neural Network (PNN) represents the physical parameters of the reduced grid and a Virtual Neural Network (VNN) approximates the noise. To find the proper output boundary, we first restrict the PNN using sparse regularization. Simultaneously, we encourage PNN to approximate the final output with priority, indirectly restricting the output of VNN. We show that such a mechanism can be easily achieved via a skip-connection (He et al., 2016b). Secondly, to achieve physical knowledge maximization, the output of PNN should be independent of the noise output of the VNN. Thus, we propose an adversarial learning scheme with Embedding Neural Networks (ENNs) to extract similar features from outputs of PNN and VNN. Adversarially, training PNN and VNN leads to maximized output dissimilarity.

Notably, such an idea is the reverse thinking of traditional deep learning-based adversarial learning that seeks maximal similarity between two targets (e.g., feature distributions). Namely, we aim to find the maximized feature dissimilarity of two NNs. Under this condition, classical training loss like binary loss in Generative Adversarial Networks (GANs) easily suffers instability. Thus, we address the problem using the similarity and dissimilarity measures of contrastive learning (Hadsell et al., 2006). Finally, we conduct extensive experiments over various systems to demonstrate the much better performance of our model compared to other state-of-the-art methods.

## 2 RELATED WORK

**Physical System Identification**. Physical system identification is a central topic for modern physical systems, especially in the edge areas. The target is to identify system governing equations using sensor measurements (Li & Weng, 2021). While the problem is similar to learning underlying equations from data, the key challenge for physical system identification is the incomplete measurement availability due to the sensor cost in a wide area of the system. Thus, traditional methods of learning equations using symbolic regression (Schmidt & Lipson, 2009; Petersen, 2019) can easily suffer overfitting. Then, (Brunton et al., 2016c; Champion et al., 2019) assume that the system is sparse and utilize Least Absolute Shrinkage and Selection Operator (LASSO) regularization to restrict physical equation representation and avoid overfitting. This regularization, however, causes inaccuracy to learn physical parameters due to the penalty term.

Thus, (Li & Weng, 2021) proposes a deep-shallow architecture to learn physics with a shallow neural network as well as approximating the hidden components with a deep neural network. However, the DNN model easily suffers from local optima, making the output boundary between the shallow neural network and the DNN deviate from the true boundary. Consequently, the recovery accuracy of the physical parameters in the shallow neural network deteriorates. To restrict the output boundary, (Yin et al., 2020) also assumes physical and augmented neural networks and propose a $l_2$ norm to regularize the parameters of the augmented NN. However, such regularization may easily cause overfitting of the physical NN. Finally, (Takeishi & Kalousis, 2021) can restricts both the physical and the virtual NNs. However, their regularization requires strict assumptions on variable distributions and the prior physical knowledge, which may not be available for general physical systems.

**Adversarial Learning**. Adversarial learning is a popular approach for training two adversarial components in a NN. Primarily, the mini-max game training helps to achieve the Nash equilibrium for optimal policy. For example, Generative Adversarial Network (GAN) (Goodfellow et al., 2014; Sauder & Sievers, 2019) utilizes a generator to generate fake data and, adversarially, utilize a discriminator to distinguish between the fake data and the true data. The optimal status is to enable the generator to accurately approximate the distribution of the true data. Such an idea is further utilized in various machine learning domains like domain adaptation (Ganin et al., 2016) to extract similar features between two domains and disentangled representation learning (Tran et al., 2017).

**Contrastive Learning**. Contrastive learning seeks representation with a minimal distance of similar samples and maximal distance of dissimilar samples (Hadsell et al., 2006). Usually, the distance is measured in an embedding space to seek the best embedding. The elaborated contrastive loss can thus guarantee stable training due to the soft comparisons between the positive and the negative samples. Such a technique is widely utilized in image embedding (Park et al., 2020), feature clustering (Li et al., 2021c), text recognition (Aberdam et al., 2021), etc.

## 3 METHODS

### 3.1 PROBLEM FORMULATION

In this paper. we study physical systems that can be modeled as a directed weighted graph $\mathcal{G} = \{\mathcal{V}, \mathcal{E}\}$, where $\mathcal{V}$ represents the vertex (node) set, and $\mathcal{E} \subseteq \mathcal{V} \times \mathcal{V}$ represents the edge set. All nodes in $\mathcal{V}$ have physical variables $\boldsymbol{x}$ and $\boldsymbol{y}$, where $\boldsymbol{x}$ represents the system state variables, and $\boldsymbol{y}$ represents the system net outputs. Then, the system equations can be written as $\boldsymbol{y} = f(\boldsymbol{x})$. For example, in electric grids, $\boldsymbol{x}$ denotes the voltage phasor, and $\boldsymbol{y}$ denotes the net power (i.e., the power consumption minus the power injection). Then, $f$ can represent the power flow equation (Yu et al., 2017). Parameters in $f$ are usually (partially) unknown to us due to the growing system size, evolving system properties, events, and maintenance, etc. Thus, it's essential to estimate parameters of $f$ using data of $\boldsymbol{x}$ and $\boldsymbol{y}$.

However, not all nodes in $\mathcal{V}$ can be equipped with sensors due to the sensor cost. Thus, we denote $\mathcal{V} = \mathcal{O} \cup \mathcal{U}$, where $\mathcal{O}$ represents the observable nodes, and $\mathcal{U}$ represents the unobservable nodes. The physical function can then be written as $[\boldsymbol{y}_{\mathcal{O}}, \boldsymbol{y}_{\mathcal{U}}]^{\top} = f\big([\boldsymbol{x}_{\mathcal{O}}, \boldsymbol{x}_{\mathcal{U}}]^{\top}\big)$, where $\top$ is the transpose operation. Subsequently, we denote the observed data samples as $\{\boldsymbol{x}_{\mathcal{O}}^n\}_{n=1}^N$ and $\{\boldsymbol{y}_{\mathcal{O}}^n\}_{n=1}^N$, where $N$ is the number of samples. Based on the above measurements, we aim to find an accurate mapping $g_{\boldsymbol{\theta}}$ such that $\boldsymbol{y}_{\mathcal{O}} \approx g_{\boldsymbol{\theta}}(\boldsymbol{x}_{\mathcal{O}})$. Further, a subset of the parameters $\boldsymbol{\theta}_p \subset \boldsymbol{\theta}$ should represent the physical parameters in $f$ as many as possible.

The missing quantities of $\boldsymbol{x}_{\mathcal{U}}$ and $\boldsymbol{y}_{\mathcal{U}}$ make the problem challenging as the under-estimate or over-estimate of $\boldsymbol{x}_{\mathcal{U}}$ and $\boldsymbol{y}_{\mathcal{U}}$ will easily cause the inaccurate results of $\boldsymbol{\theta}_p$ and further hurt the generalization ability of $g_{\boldsymbol{\theta}}$. Subsequently, this causes negative impacts on the downstream tasks like system resource optimization, reliability evaluation, and optimal expansion.

To solve the above problem, we propose twin NNs to approximate both of them. As shown in Fig. 1, the twin NNs try to map from $\boldsymbol{x}_{\mathcal{O}}$ to $\boldsymbol{y}_{\mathcal{O}}$ with physics consistency, where Physical Neural Network (PNN) learns physical parameters and outputs physical quantities and Virtual Neural Network (VNN) approximates the remaining quantities, i.e., the uncertainties and hidden measurements. Thus, they jointly contribute to the final output $\boldsymbol{y}_{\mathcal{O}}$.

Before illustrating the design details, we first introduce a moderate assumption for the system prior. Specifically, we assume the physical bases $\boldsymbol{z}$ are known in the physical equation. Namely, there is a mapping $\boldsymbol{z} = \phi(\boldsymbol{x})$ such that $\boldsymbol{y} = A\boldsymbol{z}$, where $A$ is a constant matrix representing system parameters. Then, all the system non-linearity is incorporated in $\phi$ and $A$ contains all physical information and needs to be estimated. If we know all nodes' measurements, a linear regression can help to identify $A$. However, we only have measurements of $\boldsymbol{x}_{\mathcal{O}}$ and $\boldsymbol{y}_{\mathcal{O}}$. Correspondingly, we denote $\boldsymbol{z} = [\boldsymbol{z}_{\mathcal{O}}, \boldsymbol{z}_{\mathcal{U}}]^{\top}$, where $\boldsymbol{z}_{\mathcal{O}}$ are physical bases purely calculated via $\boldsymbol{x}_{\mathcal{O}}$ and $\boldsymbol{z}_{\mathcal{U}}$ are the bases calculated via $\boldsymbol{x}_{\mathcal{U}}$ or $[\boldsymbol{x}_{\mathcal{O}}, \boldsymbol{x}_{\mathcal{U}}]^{\top}$. Thus, we fix the parameters from $\boldsymbol{x}_{\mathcal{O}}$ to $\boldsymbol{z}_{\mathcal{O}}$ and place the learnable parameters in the mapping from $\boldsymbol{z}_{\mathcal{O}}$ to $\boldsymbol{y}_{\mathcal{O}}$. Namely, we build a physical library, as shown in the left part of Fig. 1. Note that for some complex systems, the mapping $\phi$ may not be known, and we may demand symbolic regression-based methods to generate the base symbols first (Petersen, 2019). However, in this paper, we focus on the issue of incomplete system observability with the prior knowledge of $\phi$, e.g., the quadratic and sinusoidal functions in an electric grid.

### 3.2 PHYSICAL NEURAL NETWORK (PNN): LEARN PHYSICS WITH SPARSE REGULARIZATION

Due to the linear relationship between $\boldsymbol{z}_{\mathcal{O}}$ and $\boldsymbol{y}_{\mathcal{O}}$, we introduce a linear mapping $g_p$ such that $\boldsymbol{h}_1 = g_p(\boldsymbol{z}_{\mathcal{O}}) = W_1 \boldsymbol{z}_{\mathcal{O}}$ in PNN. Usually, the system structure is sparse (Brunton et al., 2016c; Champion et al., 2019). To maintain the physical consistency, we enforce a sparse regularization, i.e., the LASSO term, to guarantee sparse connections of the system. Specifically, the LASSO regularization is created as $||W_1||_1 = \sum_{i,j} |W_1[i,j]|$, where $W_1[i,j]$ is the $(i,j)^{th}$ element in $W_1$.

### 3.3 VIRTUAL NEURAL NETWORK (VNN): UNIVERSAL APPROXIMATION UNDER PHYSICAL SUPERIORITY

After showing the regularized PNN, this section will explain the modeling and benefits of VNN. The goal for VNN is to (1) approximate the remaining quantities, and (2) avoid deteriorating the output of PNN. Namely, we want to find a boundary to let PNN and VNN output properly. For the first goal, we build VNN as a fully connected neural network with multiple layers, which has a high capacity to approximate the remaining quantities. Specifically, let the mapping of the VNN be $g_v$, and we have $\boldsymbol{h}_{M+1} = g_v(\boldsymbol{h}_1)$, where $M$ is the number of layers for VNN.

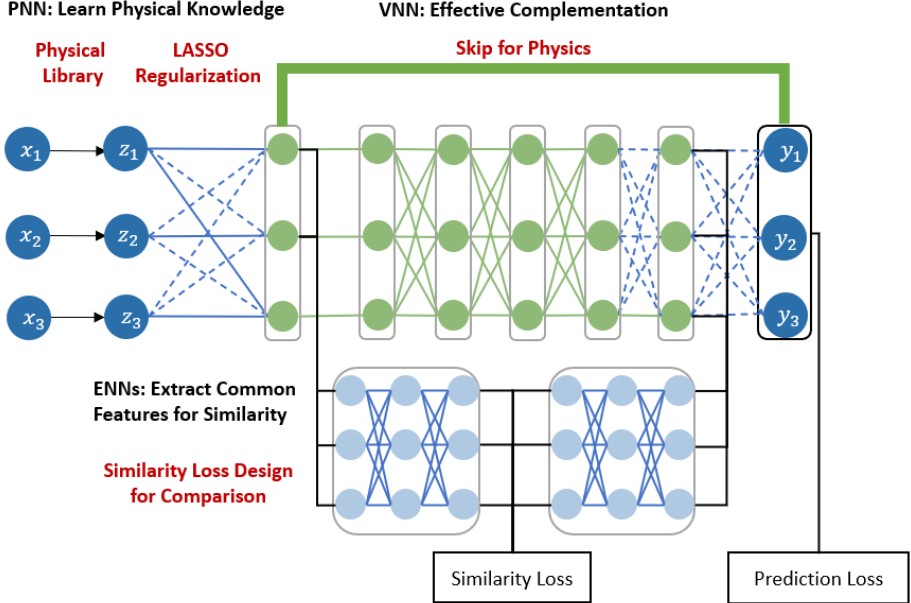

Figure 1: The structure of the proposed twin NNs.

.

For the second goal of finding the boundary, we need to restrict the output of VNN and encourage PNN to output as much as possible. On the other hand, we do not worry about the overfitting of the PNN as PNN has been strictly regularized via LASSO regularization to guarantee physical consistency. Thus, we employ a skip-connection from the output of PNN directly to the final output (He et al., 2016a). Mathematically, we have: $\boldsymbol{y}_{\mathcal{O}} = \boldsymbol{h}_1 + g_v(\boldsymbol{h}_1) = g_p(\boldsymbol{z}_{\mathcal{O}}) + g_v\big(g_p(\boldsymbol{z}_{\mathcal{O}})\big)$.

The skip-connection further ensures the possibility to create physical superiority, i.e., to encourage PNN to output as much as possible. Namely, we encourage $g_p(\boldsymbol{z}_{\mathcal{O}})$ to directly approximate $\boldsymbol{y}_{\mathcal{O}}$. In general, we have the following objective for the twin NNs.

$$\min \mathbb{E}\left[\Big(\boldsymbol{y}_{\mathcal{O}} - g_p(\boldsymbol{z}_{\mathcal{O}}) - g_v\big(g_p(\boldsymbol{z}_{\mathcal{O}})\big)\Big)^2\right] + \gamma \mathbb{E}\left[\big(\boldsymbol{y}_{\mathcal{O}} - g_p(\boldsymbol{z}_{\mathcal{O}})\big)^2\right] + \lambda||W_1||_1, \qquad (1)$$

where $\gamma$ and $\lambda$ are positive constants for the penalty terms.

### 3.4 MINI-MAX GAME TO ADVERSARIALLY TRAIN TWIN NNS FOR MAXIMIZED PHYSICAL KNOWLEDGE

In this subsection, we propose the Adversarial Twin NNs (ATN) based on the above twin NNs. Primarily, we show that ATN helps recover maximized physics, which is much better than previous work for physical system modeling.

As we mentioned in the Introduction Section, even if the boundary between the observed and unobserved measurements is successfully found, the physics may not be maximized. Essentially, we aim to find a reduced grid to govern the model and operation of observed measurements. Mathematically, we can treat the hidden components as $\boldsymbol{z}_{\mathcal{U}} = B\boldsymbol{z}_{\mathcal{O}} + \boldsymbol{e}$, where $B$ is another constant matrix for the linear representation of the unobservable parts using the observable parts, and $\boldsymbol{e}$ is the noise term.

Based on the above formulation, the maximized physics can only be achieved when $g_p(\boldsymbol{z}_{\mathcal{O}})$ represents the linear relationships from both $\boldsymbol{z}_{\mathcal{O}}$ and $\boldsymbol{z}_{\mathcal{U}} \approx B\boldsymbol{z}_{\mathcal{O}}$, and $g_v(\boldsymbol{z}_{\mathcal{O}})$ represents the output transformed from the noise term $\boldsymbol{e}$. This inspires a clear probabilistic distance between $g_p(\boldsymbol{z}_{\mathcal{O}})$ and $g_v(\boldsymbol{z}_{\mathcal{O}})$. Specifically, we aim to maximize the distribution dissimilarity between $g_p(\boldsymbol{z}_{\mathcal{O}})$ and $g_v(\boldsymbol{z}_{\mathcal{O}})$, thus achieving maximized physics. As distribution dissimilarity can hardly be explicitly

written in a neural network framework, we propose an adversarial learning framework to achieve the goal.

Specifically, we intend to *drive the training process to gradually convert physical knowledge from VNN to PNN,* which needs to measure and transfer the common knowledge. Thus, we utilize a third neural network, Embedding Neural Network (ENN), to transform the output of PNN and VNN into embedding vectors with large common knowledge, i.e., similarity of embedding distributions. On the other hand, our PNN and VNN should dynamically adjust the parameters to achieve minimal similarity. In general, the complete framework is shown in Fig. 1, where two ENNs convert the original output of PNN and VNN for the similarity loss in a mini-max game. Since we have the given output pair from PNN and VNN, the task of finding the similarity of two embeddings can be formalized as a contrastive learning framework (Hadsell et al., 2006). Thus, we define the similarity distance as follows:

$$D\Big(g_{e_1}\big(g_p(\boldsymbol{z}_{\mathcal{O}})\big), g_{e_2}\big(g_v(g_p(\boldsymbol{z}_{\mathcal{O}}))\big)\Big) = \sqrt{\mathbb{E}\Big[\big(g_{e_1}\big(g_p(\boldsymbol{z}_{\mathcal{O}})\big) - g_{e_2}\big(g_v(g_p(\boldsymbol{z}_{\mathcal{O}}))\big)\big)^2\Big]}, \quad (2)$$

where $g_{e_1}$ and $g_{e_2}$ represent the mappings of two ENNs, respectively. Namely, we utilize the Euclidean distance in the embedding space to measure the distribution similarity. Based on Equation (2), we subsequently define the similarity loss for training as follows:

$$
\begin{aligned}
L(\Theta_e, \Theta_p, \Theta_v, Y) = {} & (1 - Y)D^2\Big(g_{e_1}\big(g_p(\boldsymbol{z}_{\mathcal{O}})\big), g_{e_2}\big(g_v(g_p(\boldsymbol{z}_{\mathcal{O}}))\big)\Big) \\
& + Y\Big(\max\{0, m - D\Big(g_{e_1}\big(g_p(\boldsymbol{z}_{\mathcal{O}})\big), g_{e_2}\big(g_v(g_p(\boldsymbol{z}_{\mathcal{O}}))\big)\Big)\}\Big)^2,
\end{aligned}
\tag{3}
$$

where $\Theta_e$ represent the set of parameters in $g_{e_1}$ and $g_{e_2}$, and $\Theta_v$ and $\Theta_p$ represent the parameter sets for PNN and VNN, respectively. $Y$ is a scalar to indicate if the sample pair is similar ($Y = 0$) or dissimilar ($Y = 1$), and $m$ is a threshold to achieve stability in contrastive learning.

With the loss function above, we derive the the value function for the mini-max game between PNN and VNN:

$$
\begin{aligned}
\min_{\Theta_e} \max_{\Theta_v, \Theta_p} V(\Theta_e, \Theta_p, \Theta_v) = {} & L(\Theta_e, \Theta_p, \Theta_v, Y) - \gamma\mathbb{E}[(\boldsymbol{y}_{\mathcal{O}} - g_p(\boldsymbol{z}_1))^2] \\
& - \mathbb{E}[(\boldsymbol{y}_{\mathcal{O}} - g_p(\boldsymbol{z}_1) - g_v(g_p(\boldsymbol{z}_1))^2] - \lambda||W_1||_1.
\end{aligned}
\tag{4}
$$

During the training, we denote $Y = 0$ for the maximization process as we want to train PNN and VNN to maximize the distance in Equation (2). Conversely, we denote $Y = 1$ in the minimization process to minimize the defined distance. The adversarial training process finally leads to the maximized dissimilarity between outputs of PNN and VNN, which implies that the physical knowledge is maximally represented in PNN. Since we do the adversarial training, we denote our model as Adversarial Twin NNs (ATN).

## 4 EXPERIMENTS

To extensively verify the performances of ATN, we elaborate our models on different datasets from different physical systems. Meanwhile, we consider testing under different system observability levels, i.e., the ratio of the number of observed nodes to the total nodes, in the set $\{0.2, 0.4, 0.6, 0.8\}$ and different Signal-to-Noise-Ratios (SNRs) in the set $\{60, 80, 100, 120\}$ to mimic the realistic conditions. Then, we employ 8000 samples and conduct 5-fold cross-validation to obtain the average results for demonstration. For all the NN training, we use Adam optimizer with a learning rate hyperparameter set $\{0.001, 0.0002, 0.00005\}$, and momentum parameters $\beta_1 = 0.5, \beta_2 = 0.999$ to train 200 epochs for each experiment. All the experiments are completed with a computer equipped with Intel(R) Core(TM) i7-9700k CPU and Nvidia Geforce RTX 2080Ti GPU. The detailed experimental settings are described as follows.

### 4.1 DATASETS

**Synthetic Data**. We first utilize synthetic datasets for demonstration. Specifically, we propose a 4-node system such that:

$$
\begin{aligned}
& y_1 = x_1^2 + cos(x_2), y_2 = x_2^3 + x_1^2, \\
& y_3 = x_3 - cos(x_4), y_4 = x_4^2 + x_1.
\end{aligned}
\tag{5}
$$

The vector variable $x$ is simulated using uniform distribution $U(0, 1)$, and we construct the set $z$ based on the physical mappings (e.g., square or cosinusoidal.) in Equation (5).

**IEEE Power Systems and PJM Load Data**. There are standard power system models in IEEE, including the grid nodes, edges, and attributes. Especially, we can download the model files and the simulation platform in MATPOWER (MATPOWER community, 2020). In this experiment, we incorporate IEEE 85-node systems for testing. To conduct the simulation, we further employ real-world power consumptions in PJM Interconnection LLC (PJM) data (PJM Interconnection LLC, 2018). The load files contain hourly electricity consumption in 2017 for the PJM RTO regions. With the above data, MATPOWER produces the system states of voltage phasor to represent the system input $x$. In the meantime, the nodal net power represents the system output $y$ for training. In the power system dataset, we have $\phi = [x^2, \sin x]$.

**Mass-damper system data**. We can utilize the following equation to represent the governing law of the mass-damper system: $\dot{q} = -DRD^\top M^{-1}q$, where $q$ is the vector of momenta of the masses, $D$ is the incidence matrix of the graph, $R$ is the diagonal matrix of the damping coefficients of the damper attached to the edges, and $M$ is the diagonal mass matrix (van der Schaft, 2017). Using MATLAB, we simulate the dynamic process of the mass-damper system with 10 nodes for testing. In the mass damper system dataset, we have $\phi = [\frac{dx}{dt}, x]$.

**UF sparse matrix-based system**. We can find a large set of sparse matrix-based networks in the UF sparse matrix systems (Texas A&M University, 2011). In this experiment, we utilize the 274-node system to test. For the convenience of later presentation, we simply utilize 4-, 10-, 85-, and 274-node systems to denote the above datasets. In the UF sparse matrix dataset, we have $\phi = [x]$.

## 4.2 Benchmark Models

To validate the performance of learning physical system representation, the following benchmark methods from related literature are used.

- Deep Residual network (**Resnet**). Similar to ATN, Resnet (He et al., 2016b) also utilizes the shortcut design to pass the representation of shallow layers to deeper layers. Such a design helps Resnet to achieve excellent performances over various machine learning tasks (Guo & Du, 2019; Ma et al., 2020).

- Sparse Identification of Nonlinear Dynamics (**SINDy**) (Brunton et al., 2016a). SINDy method conducts a sparse regression to recover the physical parameters. Especially, in our paper, we assume the physical bases are known. Thus, the essence of SINDy is to utilize a LASSO regularization to select correct bases from the built-in library.

- Deep Symbolic Regression (**DSR**) (Petersen et al., 2021). DSR employs a risk-seeking policy gradient to train a reinforcement learning agent for symbolic regression.

- Equation Learner (**EQL$^{\div}$**) (Sahoo et al., 2018). EQL$^{\div}$ treat the physical bases as activation functions in a neural network. Also, the operators like summation, multiplication, and division are included in the neural network. Finally, the target neural network includes sparsity constraints for symbol selections.

- Physics-Consistent Neural Network (**PCNN**) (Li & Weng, 2021). PCNN has a deep-shallow architecture to represent the physical model, which is similar to our ATN. However, PCNN doesn't include a skip-connection for physical superiority, and mini-max game for training.

In this paper, the hyper-parameters of NNs (i.e., ATN, Resnet, DSR, EQL$^{\div}$, and PCNN) are selected via cross-validation for the best performance.

## 4.3 Model evaluation

We propose the following metrics to evaluate the generalizability and the parameter estimation results of different models.

**Generalizability**. We utilize the Mean square error (MSE) of the testing set to evaluate the model performance of predicting $y_\mathcal{O}$.

**Parameter estimation**. To evaluate the parameter estimation performance, we utilize the so-called normalized Total Vector Error ($nTVE$) (Li et al., 2021b) to evaluate the difference between the estimated $\hat{L}$ and the true physical system parameter matrix $L$:

$$nTVE(\%) = 100 \times \frac{||\hat{L} - L||_2}{||L||_2}.$$

### 4.4 MODEL GENERALIZABILITY: ATN ACHIEVES THE LOWEST MSE IN ALL CASES

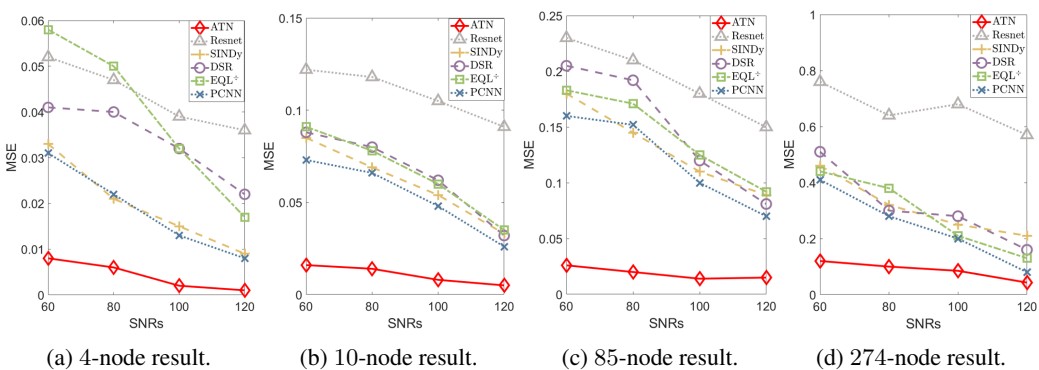

(a) 4-node result.    (b) 10-node result.    (c) 85-node result.    (d) 274-node result.

Figure 2: The results of testing MSE for different physical systems.

Firstly, we evaluate the model operational performance via generalizability for different grids and under different SNR levels. Specifically, we fix the number of observability to be $0.6$ and vary SNRS in the set $\{60, 80, 100, 120\}$. Fig. 2 demonstrates the testing MSE results for different physical systems. Clearly, we first observe that our ATN model (red line) achieves the lowest MSE for all the result points. This implies that ATN model is the most suitable model for physical system real-time operations. The following reasons help to explain the observation. (1) **ATN vs. Resnet**. In ATN model, we embed physical bases to learn a physical equation. However, for Resnet, no physics is learned. Due to the high generalizability of physical models in the observed regions, ATN performs much better than Resnet. (2) **ATN vs. SINDy**. ATN and SINDy have the ability to estimate the parameters of physical equations. However, SNIDy only utilizes LASSO as the regularization term, but ATN utilizes LASSO in PNN and employs another VNN to approximate the remaining quantities. Thus, SINDy typically suffers underfitting when there are unobserved values, which deteriorates the predicting performance.

(3) **ATN vs. DSR & EQL$^{\div}$**. DSR and EQL$^{\div}$ utilize symbolic regression to sparsely select symbols to form an equation that fits data. The missing values make the symbolic regression incapable of finding the true governing law. However, ATN can find governing equations for the reduced grid to best fit the data. (4) **ATN vs. PCNN**. Though PCNN also enjoys a twin structure to use shallow NN to learn physics and deep NN to approximate the remaining components, there is no design of the adversarial learning. Thus, training PCNN can't find the maximal physics of the reduced grid for optimal operations.

Then, we observe that for all methods, as the SNR increases, their MSE results decrease. This shows that the lower noise level helps all methods to obtain better accuracy. Interestingly, we find that for the synthetic system, SINDy performs much better than DSR and EQL$^{\div}$. For the power system, SINDy has a moderately better performance. Then, for the rest systems, SINDy, DSR, and EQL$^{\div}$ have similar performances. This is because the synthetic data has the most complex physical bases $(x, x^2, x^3, cos(x))$, and power system has a relative lower complexity $(x^2, sin(x))$. Then, the mass-damper system and UF sparse matrix have the lowest complexity $(x)$. Though we prepare all the physical base pools for each method, SINDy can successfully select the right base with LASSO regularization in the observed region. However, DSR and EQL$^{\div}$ with a NN structure can easily select the incorrect bases and suffer from overfitting. Finally, we find that PCNN usually performs better than SINDy, DSR, and EQL$^{\div}$. This shows that even without a mini-max game, the twin structure can help to identify the rough boundary of the physical and virtual outputs and obtain a more accurate approximation than methods without twin structure.

## 4.5 PHYSICAL PARAMETER ESTIMATION: ATN HAS THE MOST ACCURATE RESULTS

This subsection studies the physical-parameter-recovery ability for SINDy, DSR, EQL÷, and ATN methods. Especially, we fix the observability level to be $0.6$ and SNR $= 100$. **To make sure the ground-truth is doable for comparison, we make the nodal inputs independent of each other for each system. Thus, it avoids the case when the obtained reduced network is different from the ground-truth network, causing the impossibility for ATN method to verify results.**

Table 1 demonstrates the results of $nTVE(\%)$(mean $\pm$ standard deviation) for different datasets. First, the result shows that our proposed ATN reaches the best performance due to our sophisticated design of twin architecture and adversarial learning for dynamic boundary seeking. Secondly, as system size increases, SINDy, DSR, and EQL$^{\div}$ have the quick boosting of the $nTVE(\%)$, almost proportional to the system size growth. However, for the ATN method, $nTVE(\%)$ remains lower with only slight growth. This is because the increasing system size largely promotes the number of unobserved nodes, given fixed observability issues.

Thus, for SINDy, DSR, and EQL$^{\div}$, the probability of the perfect recovery is largely decreasing for each scalar function (i.e., a function to represent $\boldsymbol{y}_{\mathcal{O}}[i]$ for the $i^{th}$ element in $\boldsymbol{y}_{\mathcal{O}}$). Specifically, recovering $\boldsymbol{y}_{\mathcal{O}}[i]$ needs all the input variables in the function of $\boldsymbol{y}_{\mathcal{O}}[i]$. However, more unobservy errors for SINDy, DSR, and EQL$^{\div}$ increase largely. As for PCNN method, the rate of error increasing is not as high as SINDy, DSR, and EQL$^{\div}$ methods, which shows that the deep neural network in the PCNN can restrict the overfitting of the physical learner with joint training.

However, for our ATN method, though the chance for a perfect recovery decreases, our VNN can help identify the boundary between the observed physics and the remaining components under the mini-max game mechanism. Therefore, ATN can still have relatively small estimation errors for different systems.

Table 1: The $nTVE(\%)$(mean $\pm$ standard deviation) value for different methods in different systems.

|  | ATN | SINDy | DSR | EQL$^{\div}$ | PCNN |
|---|---|---|---|---|---|
| 4-NODE SYSTEM | **2.21 $\pm$ 0.68** | 8.67 $\pm$ 1.69 | 16.49 $\pm$ 3.81 | 14.25 $\pm$ 3.62 | 7.66 $\pm$ 2.01 |
| 10-NODE SYSTEM | **4.33 $\pm$ 2.08** | 14.99 $\pm$ 4.22 | 16.87 $\pm$ 4.35 | 17.73 $\pm$ 4.21 | 10.97 $\pm$ 4.19 |
| 85-NODE SYSTEM | **5.97 $\pm$ 2.32** | 34.17 $\pm$ 8.66 | 39.33 $\pm$ 7.46 | 42.21 $\pm$ 5.69 | 26.00 $\pm$ 6.23 |
| 274-NODE SYSTEM | **6.86 $\pm$ 2.56** | 54.70 $\pm$ 10.02 | 58.29 $\pm$ 11.25 | 52.17 $\pm$ 10.99 | 33.25 $\pm$ 8.19 |

## 4.6 ABLATION STUDY FOR DIFFERENT COMPONENTS OF ATN

In this subsection, we conduct the ablation study for tha ATN model to test the performance of different components. Especially, we remove the sparsity of PNN, skip-connection of VNN, and the minimax-training, respectively. We denote the models as ATN-SP, ATN-SK, ATN-MI, respectively. The results are shown in Table 2. Averagely, for the above three models, the $nTVE(\%)$ increases by $31.85\%$, $26.58\%$, and $20.21\%$. This shows that the three components have great impacts on learning the correct physics. Further, sparsity plays the essential role as it restricts the representation of the physics.

Table 2: The $nTVE(\%)$(mean $\pm$ standard deviation) value for the ablation study.

|  | ATN | ATN-SP | ANT-SK | ATN-MI |
|---|---|---|---|---|
| 4-NODE SYSTEM | **2.21 $\pm$ 0.68** | 3.02 $\pm$ 0.69 | 2.76 $\pm$ 0.81 | 2.60 $\pm$ 0.72 |
| 10-NODE SYSTEM | **4.33 $\pm$ 2.08** | 5.76 $\pm$ 3.11 | 5.01 $\pm$ 3.55 | 4.89 $\pm$ 4.21 |
| 85-NODE SYSTEM | **5.97 $\pm$ 2.32** | 7.01 $\pm$ 3.41 | 7.28 $\pm$ 3.16 | 7.04 $\pm$ 2.49 |
| 274-NODE SYSTEM | **6.86 $\pm$ 2.56** | 9.75 $\pm$ 3.29 | 9.47 $\pm$ 3.76 | 8.77 $\pm$ 2.89 |

### 4.7 SENSITIVITY ANALYSIS CONCERNING THE SYSTEM OBSERVABILITY AND SAMPLE NUMBERS

In this subsection, we consider the sensitivity analysis with respect to the system observability and the sample numbers. Specifically, for the 85-node power system, we vary the system observability, i.e., the ratio of the number of observed nodes to the total nodes, in the set $\{0.2, 0.4, 0.6, 0.8\}$. Secondly, we vary the sample number in $\{6000, 8000, 10000, 12000, 14000\}$ with observability ratio to be 0.6. Finally, we always fix SNR = 100 for testing.

Table 3 illustrates the results with respect to different observability. In general, as system observability increases, all methods have a better prediction accuracy, and ATN achieves the best performance for different scenarios. Further, with system observability level increasing from 0.2 to 0.8, MSEs of ATN, Resnet, SINDy, DSR, EQL$^{\div}$, and PCNN have a relative decrease of $80.70\%, 41.37\%, 52.94\%, 66.67\%, 63.64\%, 60\%$. It shows that ATN can make the most use of the new information from newly placed meters and achieve the most relative reduction.

Table 3: The MSE value for different methods with varying system observabilities.

| SYSTEM OBSERVABILITY | ATN | RESNET | SINDY | DSR | EQL$^{\div}$ | PCNN |
|---|---|---|---|---|---|---|
| 0.2 | **0.057** | 0.29 | 0.17 | 0.21 | 0.22 | 0.15 |
| 0.4 | **0.033** | 0.25 | 0.11 | 0.14 | 0.11 | 0.10 |
| 0.6 | **0.014** | 0.18 | 0.11 | 0.12 | 0.13 | 0.10 |
| 0.8 | **0.011** | 0.17 | 0.08 | 0.07 | 0.1 | 0.06 |

Table 4 demonstrates the results with respect to different sample numbers. We find that for different methods, when the sample number is larger than 10000, the MSE value will almost keep stable. This shows that 10000 is a corner point for the physical model learning in the 85-bus power system. Considering the high-resolution (e.g., 30 samples per second) of the Phasor Measurement Unit (PMU) (Li et al., 2021a), 10000 is an acceptable sample number for realistic training.

Table 4: The MSE value for different methods with varying training data samples.

| SAMPLE NUMBER | ATN | RESNET | SINDY | DSR | EQL$^{\div}$ | PCNN |
|---|---|---|---|---|---|---|
| 6000 | **0.022** | 0.23 | 0.15 | 0.16 | 0.14 | 0.12 |
| 8000 | **0.014** | 0.18 | 0.11 | 0.12 | 0.13 | 0.10 |
| 10000 | **0.011** | 0.15 | 0.08 | 0.11 | 0.11 | 0.08 |
| 12000 | **0.009** | 0.14 | 0.06 | 0.12 | 0.1 | 0.06 |
| 14000 | **0.010** | 0.14 | 0.05 | 0.11 | 0.1 | 0.06 |

## 5 CONCLUSION AND FUTURE WORK

To accurately model modern physical systems with incomplete system observability, we propose a twin structure of the physical neural network (PNN) and virtual neural network (VNN) to simultaneously learn physics and approximate the remaining components. Different from traditional methods, our twin structure has critical designs to find the proper output boundary between PNN and VNN: (1) we utilize sparse regularization to maintain physical consistency of PNN output, (2) we employ skip-connections to guarantee the output of the physical superiority, thus implicitly restricting the boundary of the VNN, and (3) we propose an adversarial learning scheme to adjust the fine-grained boundary for maximized physical learning. Specifically, for (3), we introduce embedding neural networks (ENNs) to extract similar features from outputs of PNN and VNN. Adversarially, training PNN and VNN leads to maximized output dissimilarity, leading to maximized physics. To guarantee the training stability, we employ the contrastive learning loss with excellent stability properties. Finally, we conduct extensive experiments over various systems to demonstrate the best performance of our model compared to other methods. For the future work, we will refine the design of PNN to tackle more complex physical systems with unknown physical bases.

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

You may include other additional sections here.

