# OpenReview forum: "Adversarial twin neural networks: maximizing physics recovery for physical system"
_ICLR.cc/2022/Conference — ICLR 2022 Submitted_

### Official Review · Reviewer_8dT9 · 2021-10-31

**Correctness:** 3
**Technical Novelty And Significance:** 2
**Empirical Novelty And Significance:** 2
**Recommendation:** 5
**Confidence:** 4

**Main Review:**

The motivation and the base problem setting are understandable, and the experiments are done on multiple datasets. However, I found not a few contradicting or unclear statements. Also, the technical novelty is not that significant. Experiments lack important ablation studies. These facts make it difficult to recommend acceptance of the current paper.

Below, I write detailed comments in an unordered way; in my intention, points (2), (3), (7), (8), (9), (10), (11), (12), (13), (14), (15), and (16) are questions, and the others are not necessarily questions (anyway, please correct if something is wrong).

(1) The beginning of the problem formulation in section 3.1 is misleading. Now it says:

> A physical system can be modeled as a directed weighted graph ...

This sounds a bit weird because many physical systems are not usually modeled as a directed graph. Instead, it would be more appropriate to say that you handle specific types of physical systems that are modeled as directed weighted graphs.

(2) Seemingly, at the layer of the problem setting, the main difference from [Li & Weng, 2021] is the presence of a possibly nonlinear function $f$ between $x$ and $y$. Meanwhile, such a difference is actually not valid because now the prior knowledge of the nonlinear functions used in $f$ is assumed as $\phi$. Consequently, in my understanding, there is no effective difference between the problem setting of the current paper and that of [Li & Weng, 2021]. I would like to hear if this understanding is correct or not. Please note that I understand the methods in both papers are more or less different; here I question the difference of the problem setting.

(3) What are "the remaining quantities" that appear at the second line of section 3.3? Do they refer to $x$'s (or $z$'s) on unobserved nodes?

(4) I fully agree with the need for restricting the output of VNN and encouraging PNN to output as much as possible. In fact, similar ideas have been already investigated in the context of physical system learning in literature such as [Yin et al., 2021] and [Takeishi & Kalousis, 2021].

[Yin et al., 2021] Yin et al., Augmenting Physical Models with Deep Networks for Complex Dynamics Forecasting, in ICLR 2021.

[Takeishi & Kalousis, 2021] Takeishi & Kalousis, Physics-Integrated Variational Autoencoders for Robust and Interpretable Generative Modeling, arXiv:2102.13156.

(5) In section 3.3, the skip-connection of the two neural nets is "proposed," while it is just a frequent practice.

(6) The regularization in equation (1) to encourage $g_p$ to directly approximate $y$ is reasonable. I note that a very similar idea was adopted in [Yin et al. 2021], where they regularize the norm of the non-physics part (i.e., $g_v$ if the notation is aligned to the current paper). I think the two approaches, i.e., minimizing $\Vert y - g_p(z) \Vert$ and minimizing $\Vert g_v(g_p(z)) \Vert$, are almost the same thing eventually after fitting gets enough good. However, in the early phase of training, since the models are not well fitted yet, perhaps there might be some practical difference.

(7) Does $y$ without subscripts in equation (1) mean $[y_\mathcal{O}; y_\mathcal{U}]$, similarly to $z$ and $x$? Maybe I missed something but could not find an exact definition.

(8) The most puzzling thing is that in section 3.4, $g_v$ suddenly starts to take $z_\mathcal{O}$ directly as an argument, whereas it used to take $g_p(z_\mathcal{O})$ as an argument in section 3.3. What are the definitions of the domain and the range of the functions around here?

(9) I could not see the point of introducing matrix $B$ in section 3.4. Does it appear somewhere in the model?

(10) What is "large common knowledge", at the fourth line of the fourth paragraph of section 3.4?

(11) What is the relation between the proposed adversarial loss and the well-known Wasserstein distance based on the Kantorovich-Rubinstein duality (see, e.g., the paper of Wasserstein GAN)? Do you have any observations on the advantages of the proposed loss, compared to the Wasserstein distance?

(12) What is the actual use of $Y$ in equation (3)? In my understanding, the idea here is to maximize dissimilarity between $g_p(z)$ and $g_v(z)$ for each $z$, and thus $Y=1$ is always used. Then, there is no need to consider the case of $Y=0$. Is this correct?

(13) What $\phi$ exactly was given to the proposed method in each experiment?

(14) In the experiment, are the baseline methods also given the knowledge of the base functions, i.e., $\phi$? If not, then the comparison is not really fair.

(15) The authors say that one of the baselines, PCNN, "doesn't include a clear design of LASSO regularization." However, in my understanding, [Li & Weng 2021] use lasso for initializing the parameters for fully observed nodes. Do you also use the same lasso-based initialization for PCNN?

(16) Seemingly, both the reported test results and the hyperparameter selection are based on cross-validation. Are there two different (and maybe nested) cross-validation? This is an important question. I would like to know the difference between the cross-validation for hyperparameter selection and that for generating the final test results.

(17) There are no ablation studies on the effect of the proposed regularizers (in equations (1) and (3)), which makes it difficult to assess the goodness of the proposed method.

(18) Twin structures of neural networks are mentioned in section 2 and some other occasions, but I could not understand why the proposed method also falls in this category, because nothing seems to be twin in the proposed method (PNN is linear, and VNN is nonlinear).

Below, minor points.

Unnecessary "and" slips in at the third-last line of page 1.

"Target is the identify ..." (the first line of page 2)

$z=[z_1, z_2]^\top$? (the lower half of page 3)

"lower nose level" (the seventh line of page 8)

**Summary Of The Paper:**

The task of identifying a physical system on a graph is addressed. While the main part of the to-be-estimated model is assumed to be linear, the proposed model needs a nonlinear part (which is modeled by a neural net) due to the presence of unobserved nodes. The authors use a combination of a sparse linear model and a neural net, which is basically the same as the model in [Li & Weng, KDD 2021]. They consider some regularization terms to maximize the use of the physics part of the model. They examine the performance of the proposed method on several datasets.

**Summary Of The Review:**

Technical novelty is limited, and there is no ablation study on the exact proposal of the current paper. Hence, I can hardly recommend acceptance in the current form of the paper.

---

> ### Author Response · Authors · 2021-11-23
> **Response to Reviewer 3**
>
> A1: Yes, we follow your suggestion and change the statement to “In this paper, we handle physical systems that can be modeled as a directed weighted graph." A2: Yes, the problem setting is the same. However, our paper doesn't need to be restricted to the specific physical systems with symmetric Laplacian matrix as the system parameter matrix. Further, for PCNN, the training may be reinitialized multiple times to obtain a good result. The reason is pointed out in our paper: the direct training can't guarantee the correct boundary between the physical output (e.g., the output of PNN) and the virtual output (e.g., the output of VNN). Thus, in this paper, we propose to utilize a mini-max game to solve this problem. A3: Yes, you are right. The remaining quantities refer to the hidden measurements. A4: For Yin's paper, please see the comparison in A3 for Reviewer 1.  As for Takeishi's paper, the authors also decompose the learning model into the physical and the virtual parts. Further, they also try to find the correct output boundary by regularization. Specifically, they regularize the virtual NN by minimizing the discrepancy between the learning model and the "physics-only" reduced model to encourage the learning model to guarantee physics superiority. Secondly, they introduce a physics-part recognition network to restrict the use of physics. In our paper, we have similar ideas for restrictions for both PNN and VNN. However, their methods require strict assumptions on the variable distributions and the known physical knowledge. On the other hand, we don't have these assumptions and utilize simple but effective mini-max games for learning. A5: Yes, we will change "propose" to "employ". A6: I agree that the two regularization terms are very similar. However, we emphasize here only utilizing the regularization may hardly guarantee the correct estimation of the underlying physical parameters. Specifically, restricting the capacity of VNN may drive the PNN to output the uncertainties and deliver incorrect PNN parameters. Thus, we need to further restrict the PNN. Essentially, this process is to find the correct output boundary between the PNN and VNN, which is never solved before. In our paper, we employ a mini-max game to gradually uncover the boundary accurately. The game mechanism encourages the dissimilar distributions between the outputs of PNN and VNN and thus, leading to correct boundary. A7: We change $\boldsymbol{y}$ to $\boldsymbol{y_O}$ in Eqn. (1). A8: It's a typo. $g_v$ should take in $g_p(\boldsymbol{z_O})$. A9: $B$ is to point out the possible relationship between the observed and the hidden measurements. We assume it's a linear mapping $B$ for explanations. Then, the parameters learned in PNN should be a function of $A$ and $B$, representing a reduced physical system within only observed nodes. A10: The common knowledge refers to the distribution similarity here. We will add the explanations. A11: The Wasserstein distance measures the distance between two distributions while our proposed loss function is a Euclidean-based loss to measure the distance of sample pairs in contrastive learning. In our case, since we compare the distance of the output pairs between PNN and VNN, we employ the Euclidean-based loss. Further, as in the mini-max game, embedding networks are added to convert the output pairs of the NN twin to an embedding space, the proposed sample pair loss is enough to capture the distribution similarity for the output pairs. A12: our paper utilizes a mini-max training scheme to learn the representation with a maximal dissimilarity. We can't do the direct maximization because we can't directly obtain the correct embedding neural networks to find the similarity of outputs between PNN and VNN. In another word, we need a minimization process to specially train the embedding neural network and learn the similarity of the output of PNN and VNN, i.e., $Y=0$ in the minimization process. A13:  In our experiments, we set $\phi=[x^2,\sin x]$ for power systems. We add all explanations for each scenario to the paper. A14: Yes, all the baseline methods include the knowledge of base functions. A15: We follow exactly the same initialization process as the original paper in PCNN. We admit that PCNN also utilizes LASSO regularization and delete the statement about PCNN in our paper. A16: We utilize the same cross-validation to select hyparameters and generate the results. We fine-tune the parameters based on the validation accuracy and generate the results based on the testing accuracy. A17: Please see A2 for Reviewer 1. A18: The twin may not need to be symmetrical. We utilize twin here to simply indicate that the two networks are coupled to achieve the same target to minimize the total regression error. A19: We have addressed all the minor errors.

---

> > ### Comment · Reviewer_8dT9 · 2021-11-29
> > **Thanks for the response**
> >
> > Thank you so much for the authors' response. I think it clarified some of my initial concerns (some unclear statements & lack of ablation study) and thus increased the score.
> >
> > Point 4,6: As for the difference from [Yin+ 2021], I think elaborating it is highly useful for readers with relevant context. As I commented in the initial review, the practical difference is only between regularizing $\Vert y - g_p(z) \Vert$ or regularizing $g_v(g_p(z))$, but such a difference implies not a few points as you explained in the response. In the revised version it is only mentioned slightly in section 2, but I would recommend, for example, creating an independent subsection in the appendix.
> >
> > Points 11-12: I am not convinced on this issue. The response makes sense --- you need the case of $Y=0$ to make the embedding networks meaningful. But, then, how do data samples for $Y=0$ come? For $Y=0$ case, $g_p(z_\mathcal{O})$ and $g_v(g_p(z_\mathcal{O}))$ should be similar; I cannot see how such data pairs can appear, because the intention of the regularization was to make the two as dissimilar as possible. By the way, in Wasserstein distance-based objectives, the "learning good embedding network" issue is addressed by constraining / regularizing the Lipschitzness of the discriminator / critic network.
> >
> > Point 16: I am confused here... by saying " the same cross-validation to select hyperparameters and generate the results", do you mean the same *configuration* of cross-validation and different *realization* of it? If yes, then it makes sense (and should be stated so clearly in the text).
> >
> > Additional comment on related work: I think the context of *diversity in ensemble" would also relate to the idea of the authors' regularizer. E.g., in [Rame & Cord 2021], they use an adversary classifier to make the components of an ensemble be diverse.
> >
> > [Rame & Cord 2021] DICE: Diversity in Deep Ensembles via Conditional Redundancy Adversarial Estimation, ICLR 2021.

---

### Official Review · Reviewer_dem1 · 2021-11-02

**Correctness:** 3
**Technical Novelty And Significance:** 3
**Empirical Novelty And Significance:** Not applicable
**Recommendation:** 5
**Confidence:** 2

**Main Review:**

Several things need to be clarified before a proper review can be put in place:

#1. Beginning of section 3.1 The author frame the problem as a graph with node set V and edges VxV. Each node in V contains (x,y) with the relationship y = f(x). Does this mean that (x,y) are associated with each node and for different nodes there are different (x,y)?

#2. Then the assumption is made for some nodes being observed and some not observed. V = O union U. The the new symbols are introduced [y_o,y_u] = f( [x_o, x_u]). Seems to me that y_o is a set instead of variables representing one specific node. Otherwise this equation [y_o,y_u] = f( [x_o, x_u]) does not make sense. If so, then the (x,y) in my point #1 are variables, not set and here y_o is a set of variables. Please clarify this point

#3. There could be a typo here: “Correspondingly, we denote z = [z1,z2]⊤, where zO ..”  What are z1 and z2?

#4. \phi the last paragraph of section 3.1 represent the basis coefficients for some known basis functions? Are the values of \phi learned end-to-end in this network? By learning PNN?

#5. Eq(3) the variable Y. Y=0 if sample pair is similar and Y=1 otherwise. In Eq(3) we only pass in one zo, not clear where the ‘pair’ comes from. Perhaps it is due to the confusion I brought up in #2. If zo is a set of variables, then how to apply a function to it? Element wise application?


**Summary Of The Paper:**

Paper presents a method to push physical quantity to a physical neural network (PNN) and push the ‘additional quantity’ such as noise to a correction term virtual neural network (VNN). Uses a pair of embedding neural network (ENN) to achieve the above.

**Summary Of The Review:**

will do a more proper review after my queries have been addressed. Will change my score after the proper review.

---

> ### Author Response · Authors · 2021-11-23
> **Response to Reviewer 2**
>
> A1: $\boldsymbol{x}$ and $\boldsymbol{y}$ represent the system input and output for system nodes and each elements in $\boldsymbol{x}$ and $\boldsymbol{y}$ represent one nodes' measurements. A2: Yes, your point in Q1 is correct. Please see my answers in A1. A3: Yes, it's a typo. We denote $\boldsymbol{z}=[\boldsymbol{z_O},\boldsymbol{z_U}]$. A4: In this paper, we assume the physical bases $\phi$ are known. A5: As shown in Fig. 1, $g_{e_1}(g_p(\boldsymbol{z_O}))$ and $g_{e_2}(g_v(\boldsymbol{z_O}))$ represent the pair. $\boldsymbol{z_O}$ is a set of variables and the function represents a neural network.

---

> > ### Comment · Reviewer_dem1 · 2021-11-30
> > **still not clear**
> >
> > after reading the answers to my question 1 and 2, it is still not clear to me which is a set, which is a variable. symbols y_o and y_u are also confusing.

---

### Official Review · Reviewer_FaHY · 2021-11-02

**Correctness:** 2
**Technical Novelty And Significance:** 2
**Empirical Novelty And Significance:** 3
**Recommendation:** 6
**Confidence:** 3

**Main Review:**

Physical system identification is a definitely interesting topic in both the machine learning and physics communities. The authors address such an important issue and validate their proposal empirically. The proposed ATN show remarkable results in terms of model accuracy and unknown parameter estimation, with robustness regarding the sample size and observability.

However, there are some concerns that should be addressed for clear evaluation of this paper:

* Q1. The authors assume that there are physics bases $z$ that yield a linear relationship $y = Az$. In addition, they assume the unobservable base is also a linear transformation of the observable one, i.e., $z_U = B z_O + \epsilon$, where $\epsilon$ is relatively small noise. I wonder this assumption is general for arbitrary physical systems or just for fine the examples presented in this paper.

* Q2. Because the proposed architecture is complicated, I wonder all the components are truly necessary. The authors claim the usefulness of each component indirectly, by comparing other baselines (e.g., the usefulness of physics bias with comparing ResNet, that of VNN with comparing SINDy, …), However, a more systematic ablation study should be included in the paper (or Appendix if the page is not sufficient), to clearly support the necessity of the complicated ATN structure.

* Q3. It seems that the following paper is highly relevant: "Y Yin et al. Augmenting physical models with deep networks for complex dynamics forecasting. ICLR 2021." This paper also decomposes the unknown physical system as a known physics part (however whose parameters are unknown) and the neural network part that fills the remainder that cannot be explained by the physics-based one.

* Q4. It seems that the proposed ATN already knows the appropriate physics bases $z = \phi(x)$, referring to the explanation on page 3 (“we fix the parameters from $x_O$ to $z_O$”) and equations (1 – 4) (no $x$ and $\phi$ in the objectives). However, on page 7, the authors state that “ATN and SINDy can both learn physical bases”, which seems that there is a searching procedure of physics bases from a set of predefined operations. Can the authors elaborate it more clearly?

* Q5. There are some inconsistent or confusing notations in the paper, e.g., on page 3 “Correspondingly, we denote $z = [z_1, z_2]^T$, where $z_O$ are physical bases…”, on page 4, $y_O = g_p(z_O) + g_v(g_p(z_O))$ should be $y$ rather than $y_O$ (because $g_v$ is included in the expression, and consistency with (1 – 4))?. In addition, Figure 1 is not clear because all subscripts of variables are 1, 2, … rather than $O$ or $U$.




**Summary Of The Paper:**

The paper proposes ATN to model and identify physical systems. Here, the physical systems are grids consisting of measurement sensors: because the sensor configuration is not perfect, there can be unobservable parts in the grid system. The authors assume that there are physics bases that can represent the output with a linear mapping. Thus, they firstly use a linear LASSO regressor (PNN in the paper) that transforms observable physics bases to the output. The possible remainder (due to the noise or unobservable physics) is modeled via a more complicated neural network (VNN in the paper). These two networks are integrated into a single skip-connection block. They are trained by MSE, with enforcing the PNN part predict the output mostly, to exploit the bias of physics. In addition, the authors suggest the use of siamese networks that measure the distance between PNN and VNN: then an adversarial learning loss is added to guarantee the outputs of PNN and VNN to be far from each other and play their respective roles more concisely. ATN outperforms its counterparts for some physical system datasets.



**Summary Of The Review:**

Currently, I lean towards rejection due to the concerns I raised. I would like to update my evaluation after discussing with the authors and my fellow reviewers.

***
Post-rebuttal:  I appreciate the authors’ clarifications on my questions. I am happy to see the ablation study result, which can support the necessity of the proposed components more clearly.

However, the essential assumptions used in this paper ($y = A \phi(x)$ with known $\phi$ and $z_U = B z_O + \epsilon$) make the proposed model be applicable only for relatively restricted problems. It prevents me from recommending a clear acceptance for this paper. While I slightly tend to accept this paper (thus I updated the review score accordingly), I would not champion this paper for the acceptance.

---

> ### Author Response · Authors · 2021-11-23
> **Response to Reviewer 1**
>
> A1: In this paper, we assume physical bases $\phi$ are known. Then, for an arbitrary system, a linear relationship is sufficient to represent the system $\boldsymbol{y}=A\boldsymbol{z}=A\phi(\boldsymbol{x})$ and the nonlinearity only happens in $\phi$. Further, for the equation $\boldsymbol{z_U}=B\boldsymbol{z_O}+\boldsymbol{e}$, we assume there is a linear relationship between the observed and hidden quantities. This may not be generally true for all physical systems and we will elaborate on other relationships in future work.
> A2: We conduct the ablation study. Specifically, we remove the sparsity of PNN, skip-connection of VNN, and mini-max training, respectively. Then, we test the models in different systems and achieve similar results. For example, for 85-node system and compared to the complete ATN model, the mean of nTVE(\%) has an increase of 35\%, 22\%, and 18\%, respectively. This shows that the three components have a great impact on learning the correct physics. Further, sparsity plays an essential role as it restricts the representation of physics.
> A3: The authors in Yin's paper also assume physical and augmented neural networks to represent the underlying dynamical equations of the physical model. However, their method has the following disadvantages and is unqualified for our setting. (1) their design of restricting the output of augmented NN is to minimize its norm. For our dataset, this restriction easily causes overfitting of the physical NN. Specifically, the physical parameters will be misestimated to enable the physical NN to fit the part that augmented NN can't approximate. Correspondingly, in our model, we also restrict the VNN's representation using a skip connection. However, we add a mini-max learning scheme to maximize the dissimilarity between the outputs of PNN and VNN. Thus, PNN can't represent the noise that should be handled by VNN, leading to a better output boundary between PNN and VNN. (2) they consider a noiseless dataset while the noise can further deteriorate the correct boundary. (3) they only focus on low-dimensional data but don't present results for high-dimensional data.
> A4: The physical bases are assumed to be known and we need to estimate the parameters. On page 7, our claim should be ``ATN and SINDy have the ability to estimate parameters of physical equations."
> A5: We re-clarify our notations. On page 3, we denote $\boldsymbol{z}=[\boldsymbol{z_O},\boldsymbol{z_U}]$. On page 4, the notation should $\boldsymbol{y_O}$ rather than $\boldsymbol{y}$. The reason is that $\boldsymbol{x_O}$ and $\boldsymbol{y_O}$ are the observed input and output measurements of the physical system and we don't have the unobserved measurements $\boldsymbol{y_U}$ for training. Further, the unobserved measurements $g_v(g_p(\boldsymbol{z_O}))$ should still contribute to the observed measurements $\boldsymbol{y_O}$ due to the possible physical connections between unobserved nodes and observed nodes. Finally, we will modify Fig. 1 for better clarification.

---

> > ### Comment · Reviewer_FaHY · 2021-11-29
> > **Reply to the authors' response**
> >
> > I appreciate the authors’ clarifications on my questions. I am happy to see the ablation study result, which can support the necessity of the proposed components more clearly.
> >
> > However, the essential assumptions used in this paper ($y = A \phi(x)$ with known $\phi$ and $z_U = B z_O + \epsilon$) make the proposed model be applicable only for relatively restricted problems. It prevents me from recommending a clear acceptance for this paper. While I slightly tend to accept this paper (thus I updated the review score accordingly), I would not champion this paper for the acceptance.

---

### Author Response · Authors · 2021-11-23
**Response Summary**

Thanks a lot for the reviewers' questions. We have addressed all the comments. Please see the revised paper with modifications marked in blue. As for specific questions, please see our comments. Thanks for considering our paper.

---

### Decision · Program_Chairs · 2022-01-20

**Decision:**

Reject

**Comment:**

This paper addresses the identification of physical systems defined on graphs. The authors introduce the Adversarial Twin Neural Network (ATN), which consists in augmenting a simple linear model (PNN) with a virtual neural network (VNN). Some regularization terms are used to enforce maximum prediction from the PNN, and to enforce diverse outputs between PNN and VNN.

The paper initially received tree rejection recommendations. The main limitations pointed out by reviewers relate to the limited contributions, the limiting assumption of using a linear mode for PNN, the lack of positioning with respect to related works, and clarifications on experiments. The authors' rebuttal answered to some reviewers concerned: Rdem1 increased its grade from 3 to 5, and Rdem1 from 5 to 6 - although not willing to champion the paper. R8dT9, which provided a very detailed review and feedback after rebuttal still voted for rejection, especially because he was not convinced by the positioning with recent related works and the answers on experiments.

The AC's own readings confirmed the issues essentially raised by R8dT9 and other reviewers. Especially, the AC considers that:
- The contributions for driving a proper cooperation between the PNN and VNN models are weak, since it reduces to using simple skip connection and adversarial training.
- The importance of these aspects have not been analysed in depth in the revised version of the paper, neither theoretically nor experimentally: for example, the difference with respect to [Yin+ 2021] for a proper augmentation, the discussion to alternative methods for representing diversity as done in [Rame & Cord 2021], or the positioning with respect to  Wasserstein distance-based objectives.
- There remains ambiguities in the cross-validation process, which have not been addressed in the rebuttal.

Therefore, the AC recommends rejection.